# BTLA-Expressing Memory B Cells Are Associated with Belimumab-Induced Improvement in Systemic Lupus Erythematosus

**DOI:** 10.3390/ijms262311323

**Published:** 2025-11-23

**Authors:** Takuya Nishi, Kunihiro Hayakawa, Keigo Ikeda, Maki Fujishiro, Yuko Kataoka, Ken Yamaji, Kenji Takamori, Naoto Tamura, Iwao Sekigawa, Shinji Morimoto

**Affiliations:** 1Institute for Environment and Gender-Specific Medicine, Juntendo University Graduate School of Medicine, 2-1-1 Tomioka Urayasu-Shi, Chiba 279-0021, Japan; ta-nishi@juntendo.ac.jp (T.N.); khayaka@juntendo.ac.jp (K.H.); mfujishi@juntendo.ac.jp (M.F.); y-kataoka@juntendo.ac.jp (Y.K.); ktakamor@juntendo.ac.jp (K.T.); morimoto@juntendo.ac.jp (S.M.); 2Department of Internal Medicine and Rheumatology, Juntendo University Urayasu Hospital, Chiba 279-0021, Japan; 3Department of Internal Medicine and Rheumatology, School of Medicine, Juntendo University, Tokyo 113-8421, Japan; k.yamaji@juntendo.ac.jp (K.Y.); tnaoto@juntendo.ac.jp (N.T.)

**Keywords:** belimumab, systemic lupus erythematosus (SLE), B- and T-lymphocyte attenuator (BTLA), Bruton tyrosine kinase (Btk), phospholipase C gamma 2 (PLCγ2), B cell receptor signaling, B cell-activating factor (BAFF)

## Abstract

Belimumab, a fully humanized B cell-activating factor (BAFF)-targeting monoclonal antibody, inhibits autoreactive B cell survival and improves systemic lupus erythematosus (SLE) clinical outcomes. However, its administration criteria remain unclear. To establish a basis for defining these criteria, we characterized the immune cell subpopulation alterations post-belimumab treatment and elucidated the underlying mechanisms. We hypothesized that belimumab modulates specific cell subsets and investigated the post-therapy changes. Flow cytometry and correlation analysis revealed that the frequency of B- and T-lymphocyte attenuator (BTLA)^high^ memory B cells in peripheral blood and clinical improvement after belimumab treatment. Western blotting analysis of healthy control B cells revealed that BTLA engagement suppressed Bruton tyrosine kinase and phospholipase C-gamma 2 phosphorylation, which was enhanced by B cell and BAFF receptor co-stimulation. BTLA-expressing memory B cells, which positively correlate with disease improvement, possibly contributed to SLE improvement via BTLA-mediated signaling that attenuated B cell- and BAFF receptor-induced intracellular pathways. To validate these findings, we plan to further assess the effects of belimumab on BTLA expression and B cell signaling pathways in treatment-naive patients with SLE by western blotting. Collectively, our results provide a novel foundation for establish appropriate belimumab administration criteria.

## 1. Introduction

Systemic lupus erythematosus (SLE) is a systemic autoimmune disease in which genetic predispositions and environmental factors cause abnormalities in the innate and adaptive immune systems. These abnormalities contribute to autoantibody production and immune complex formation, resulting in multiple organ damage, including nephritis, and diverse symptoms. B cells play a role in SLE pathogenesis, leading to autoantibody production, inflammatory cytokine secretion, and antigen presentation. B cell-activating factor (BAFF), a member of the tumor necrosis factor (TNF) family, is a cytokine implicated in the survival and maturation of B cells. BAFF is involved in the pathogenesis of SLE [1], and its expression correlates with disease activity in SLE [2].

Belimumab, a fully humanized immunoglobulin (Ig)G1 monoclonal antibody (mAb), became the first targeted therapy for SLE in Japan at the end of 2017. Belimumab neutralizes BAFF, thereby inhibiting B cell proliferation and differentiation, regulating B cell function, and triggering autoreactive B cell apoptosis [3]. Clinically, belimumab treatment improves the disease activity and reduces the flare frequency and steroid dose [4].

Currently, belimumab is used for the following purposes: (1) To treat patients resistant to standard therapies (steroids and other immunosuppressants), (2) to maintain remission and suppress SLE relapse, and (3) to reduce steroid use [4]. Previous trials have demonstrated the efficacy of belimumab for lupus nephritis [5].

B- and T-lymphocyte attenuator (BTLA), a type I membrane protein belonging to the Ig superfamily that acts as a co-inhibitory receptor [6], is possibly involved in the pathogenesis of autoimmune diseases [7]. The ligand for BTLA is TNF receptor superfamily member 14 (also known as herpes virus entry mediator [HVEM]), a type I membrane protein belonging to the TNF receptor superfamily [8]. Ligation of BTLA by HVEM attenuates T cell activation, leading to the inhibition of T cell proliferation [9]. It also attenuates B cell activation, leading to the inhibition of B cell proliferation and downregulation of costimulatory molecules and cytokine production [10]. In the MRL/*lpr* lupus mouse model, BTLA deletion exacerbates lupus, suggesting the protective role of BTLA in SLE pathogenesis [11]. Although the precise roles of BTLA and HVEM in human SLE have not been fully elucidated, we previously showed that HVEM expression changes during the menstrual cycle in females [12]. Furthermore, by analyzing gene expression in the peripheral blood mononuclear cells (PBMCs) of females with SLE, we reported decreased HVEM expression levels in patients with active SLE compared to those in the healthy controls and patients with inactive SLE [12]. Sawaf et al. showed that the ability of BTLA to inhibit T cell activation is suppressed in SLE and that BTLA dysfunction in lupus CD4^+^ T cells is associated with the disease activity [13]. However, the roles and effects of BTLA and HVEM in the pathogenesis of human SLE remain ambiguous.

In our hospital, belimumab administration improves the clinical characteristics of most patients with SLE that is refractory to standard therapies, with only a minority failing to respond to treatment. However, the appropriate administration criteria according to the clinical context remain unclear. To identify the patient populations with good responses to belimumab and determine the exact effects of belimumab on the cell populations in patients with SLE, we conducted a longitudinal study with serial collection of peripheral blood samples for the analysis of treatment-associated changes in immune cell subsets.

We focused on the effects of belimumab treatment on BTLA and HVEM levels in the lymphocytes of patients with SLE and analyzed the associations between these immune checkpoint proteins and the clinical manifestations of SLE.

We also examined other effects of belimumab, including BAFF inhibition, to provide a strong foundation for establishing appropriate belimumab administration criteria.

## 2. Results

### 2.1. Effects of Belimumab Treatment on PBMC Subsets

PBMC subsets were analyzed via flow cytometry using samples collected before and 1, 3, 6, and 12 months after the initiation of belimumab treatment.

To evaluate the potential immunological changes, we initially examined the T cell subsets. The frequency of total T cells (CD3^+^CD19^−^) and CD4^+^ T cells remained stable throughout the observational period (Figure 1A,B). This finding is consistent with previous reports that belimumab does not significantly affect T cell subpopulations [14,15].

Next, we analyzed the B cell subsets. Previous studies have shown that belimumab reduces the B-cell populations [14,16]. Although previous studies have generally demonstrated a decrease in the frequency of total B cells (CD3^−^CD19^+^) in the lymphocyte population during the observational period, our study showed no significant change (Figure 2A). Further analysis revealed that the frequency of HVEM^high^ naive B cells (CD19^+^CD27^−^) significantly decreased at 3 and 6 months, whereas that of HVEM^high^ memory B cells (CD19^+^CD27^+^) significantly increased at the same time point (Figure 2B). Notably, BTLA^high^ naive B cell frequency significantly decreased at 3 months, BTLA^high^ memory B cell frequency tended to increase over time, and BTLA^low^ naive B cell frequency significantly increased at one month after treatment (Figure 2C).

Taken together, these data suggest that memory B cells specifically migrate from tissues into the circulating blood and that their high expression of HVEM and BTLA contributes to the suppression of pathological autoreactivity.

### 2.2. Correlations Between B Cell Subpopulations and Clinical Data

Existing research on changes in HVEM- and BTLA-expressing B cell subpopulations in patients with SLE after belimumab treatment remains limited. However, understanding these immunological dynamics can provide important insights into belimumab therapeutic mechanisms. Therefore, we analyzed the correlations between the frequencies of HVEM- and BTLA-expressing B cell subpopulations and serological biomarkers or SLE disease activity index 2000 (SLEDAI-2K) scores that changed in response to belimumab to examine their relationship with disease activity.

Correlation analysis revealed several significant associations. HVEM^high^ naive B cell frequency was significantly inversely correlated with SLEDAI-2K scores and exhibited an inverse trend with serum IgG levels (Figure 3A). Moreover, HVEM^high^ memory B cell frequency was significantly positively correlated with serum C4 levels (Figure 3B). BTLA^high^ naive B cell frequency exhibited an inverse correlation with SLEDAI-2K scores but did not show significant correlations with anti-DNA antibody titers or serum C4 levels (Figure 3C). In contrast, BTLA^low^ naive B cell frequency did not correlate significantly with SLEDAI-2K scores but was significantly positively correlated with anti-DNA antibody titers and significantly inversely correlated with serum C4 levels (Figure 3D).

BTLA^high^ memory B cell frequency was significantly positively correlated with serum C4 levels and inversely correlated with serum IgG levels (Figure 3E). Conversely, BTLA^low^ memory B cell frequency showed no significant correlation with serum C4 levels but was significantly positively correlated with serum IgG levels (Figure 3F).

These findings suggest that HVEM^high^ naive and memory B cells are associated with improvement in disease activity (Figure 3A,B). Moreover, BTLA^low^ naive B cells were associated with disease exacerbation (Figure 3D), whereas BTLA^high^ naive and memory B cells were associated with disease improvement (Figure 3C,E). Notably, belimumab treatment was associated with a trend toward an increase in BTLA^high^ memory B cell frequency (Figure 2C). Collectively, these results suggest that the increase in BTLA-expressing memory B cell frequency induced by belimumab treatment is associated with improved disease activity. These findings suggest a potential mechanism by which belimumab exerts its therapeutic effects in patients with SLE.

### 2.3. Effects of BAFF and B Cell Receptor (BCR) Stimulation on HVEM and BTLA Expression Levels on B Cells

BTLA expression levels are reduced on naive and double-negative (DN; CD27^−^IgD^−^) memory B cells in patients with SLE [17]. HVEM protein levels in B cells are also decreased in patients with SLE [18]. However, the mechanisms by which BAFF affects HVEM and BTLA levels remain unclear. To investigate the effects of BAFF stimulation on HVEM and BTLA levels in B cells, we measured the HVEM and BTLA expression levels on each B cell subpopulation of healthy controls stimulated with or without anti-human IgM and human BAFF. As shown in Appendix A, HVEM levels on the total and memory B cells increased after stimulation with anti-IgM alone or in combination with BAFF. In contrast, BTLA levels on total, naive, and memory B cells decreased upon anti-IgM stimulation with or without BAFF (Appendix A). Collectively, these results suggest that B cell activation via BCR signaling promotes HVEM upregulation and BTLA downregulation in vitro. Notably, these changes were not observed following stimulation with BAFF alone. Therefore, BAFF inhibition by belimumab may indirectly alter the proportions of HVEM- and BTLA-expressing B cell subpopulations, potentially via intracellular signaling pathways.

### 2.4. Modulation of BCR/BAFF-Induced B Cell Intracellular Signaling by BTLA

BCR, a transmembrane receptor expressed on the B cell surface, generates signals in response to antigen recognition and activates a network of downstream signaling pathways [19]. BCR engagement initiates intracellular signaling that leads to the phosphorylation of spleen tyrosine kinase (Syk), which subsequently activates Bruton tyrosine kinase (Btk) via phosphorylation [20]. Activated Btk further phosphorylates phospholipase C-gamma 2 (PLCγ2) [21], leading to the activation of multiple downstream signaling pathways crucial for B cell survival, proliferation, and differentiation.

HVEM and BTLA interaction induces an inhibitory signal that attenuates BCR-mediated activation, thereby reducing phosphorylation of the downstream signaling molecules, including Syk, PLCγ2, and other related effectors [22]. Based on these findings, we hypothesized that BAFF signaling amplifies BCR-induced phosphorylation of Syk, Btk, and PLCγ2 and that BTLA engagement suppresses this BAFF-potentiated phosphorylation cascade. To examine the mechanisms by which BTLA modulates BCR signaling proteins under BAFF stimulation, we evaluated the anti-IgM- and BAFF-induced phosphorylation of Syk, Btk, and PLCγ2 with or without BTLA engagement.

B cells from healthy donors were stimulated with various combinations of anti-IgM, BAFF, and BTLA engagement, and phosphorylation of signaling molecules was analyzed by western blotting. Co-stimulation with anti-IgM and BAFF significantly enhanced Syk phosphorylation. However, BTLA engagement did not significantly attenuate this enhanced Syk phosphorylation (Figure 4A). In contrast, enhanced phosphorylation of Btk induced by co-stimulation with anti-IgM and BAFF was significantly attenuated by BTLA engagement (Figure 4B). Similarly, co-stimulation with anti-IgM and BAFF markedly enhanced PLCγ2 phosphorylation, and this effect was significantly attenuated by BTLA engagement (Figure 4C). These findings suggest that BTLA engagement attenuates BAFF-enhanced phosphorylation of Btk and PLCγ2, with only a minimal impact on Syk phosphorylation.

### 2.5. BTLA-Mediated Regulation of Non-Canonical NF-κB Signaling in B Cells

Nuclear factor-κB (NF-κB) is a transcription factor essential for the activation, proliferation, and immune responses of B cells [23]. NF-κB activation occurs via two distinct signaling pathways: Canonical and non-canonical pathways [24,25]. BAFF signaling mainly activates the non-canonical NF-κB pathway via a mechanism involving the processing of p100 (NF-κB2) to p52 [24] and activates the canonical NF-κB pathway via a mechanism involving Btk-dependent activation of IκB kinase complexes [25]. As described above, BTLA engagement attenuates BCR signaling, which subsequently suppresses the canonical NF-κB pathway [22]. However, the specific effect of BTLA engagement on the BAFF-driven non-canonical NF-κB pathway remains unclear. To clarify this, we evaluated the conversion of p100 to p52 following anti-IgM and BAFF stimulation in the presence or absence of BTLA engagement. As shown in Appendix A, although BAFF stimulation with or without anti-IgM reduced the p100 levels, BTLA engagement tended to increase the baseline p100 levels and attenuate BAFF-induced reduction. Conversely, p52 levels tended to increase following co-stimulation with anti-IgM and BAFF, and this effect may be suppressed by BTLA engagement (Appendix A). Collectively, findings suggest that BTLA engagement could increase the p100 levels and reduce p100-to-p52 processing, indicating its potential inhibitory effect on the non-canonical NF-κB pathway regardless of anti-IgM and BAFF stimulation.

## 3. Discussion

Since its approval for SLE treatment, belimumab has been widely used, resulting in the improvement of disease activity and reduction in flares and damage accrual [26]. In the European Alliance of Associations for Rheumatology 2023 recommendations for SLE management, belimumab is positioned at the same level as immunosuppressive drugs and specifically recommended for patients unresponsive to hydroxychloroquine alone or in combination with steroids and for those unable to reduce steroid doses to below acceptable doses [27]. Several clinical trials have demonstrated the effectiveness of belimumab in treating mucocutaneous and musculoskeletal [28], active lupus nephritis [5], and high titers of anti-double-stranded DNA antibodies [14]. However, the specific patient group clinically benefitting from belimumab treatment remains unclear.

Belimumab alters B-cell subpopulations [15,29]; however, its effects on HVEM and BTLA have not yet been reported. In this study, belimumab affected the HVEM- and BTLA-expressing B cell subpopulations in patients with SLE (Figure 2B,C). Our findings suggest that changes in these subpopulations are related to belimumab therapeutic mechanisms.

HVEM serves as a ligand for BTLA [8]. The role of BTLA engagement by HVEM in the pathogenesis of autoimmune diseases has recently gained attention [30]. In T cell immunity, BTLA binds to HVEM and inhibits T cell activation and proliferation by attenuating T cell receptor signaling as well as follicular helper T cell-induced maturation of activated B cells and differentiation into antibody-producing cells [31]. In B cell immunity, HVEM–BTLA binding negatively regulates B cell activation, inhibits proliferation, and attenuates downstream molecules of BCR signaling [22]. BTLA levels on naive B cells is lower in patients with SLE than in healthy controls and inversely correlated with the levels of markers for anti-double-stranded DNA antibodies [17].

A recent study reported that BTLA expression levels on DN memory B cells are lower in patients with SLE than in healthy controls and that BTLA^low^ DN memory B cell frequency is increased in SLE, particularly in active SLE [32]. In this study, BTLA-expressing B cells may contribute to improved disease activity (Figure 3C–F). Notably, belimumab may improve SLE disease activity via BTLA^high^ memory B cells (Figure 2C and Figure 3C,E). In our study, HVEM^high^ memory B cells also increased following belimumab treatment (Figure 2B) and were associated with improved disease activity (Figure 3B). Therefore, these findings suggest that HVEM^high^ memory B cells may cooperate with BTLA^high^ memory B cells to suppress disease activity. BTLA is a negative regulator of BCR signaling; however, the effect of BAFF inhibition on BTLA remains unclear. Here, BAFF stimulation did not modulate BTLA expression on B cells (Appendix A). This suggests that belimumab would not regulate BTLA expression in vivo. Therefore, we hypothesized that the mechanism by which BAFF inhibition alters the frequency of BTLA-expressing B cells involves other indirect pathways, including intracellular signaling pathways.

Syk, a crucial player in signal integration during BCR-induced human B cell activation [33], is downregulated upon BTLA binding to HVEM [22]. However, although co-stimulation by BCR and BAFF in the presence of BTLA engagement only slightly inhibited Syk phosphorylation (Figure 4A), BTLA engagement substantially modulated the phosphorylation of downstream signaling molecules, such as Btk and PLCγ2 (Figure 4B,C), leading to the suppression of the BCR/BAFF-induced signaling pathway in this study.

Interestingly, Btk also plays a crucial role in the activation of the non-canonical NF-κB pathway. Specifically, Btk is activated in response to BCR and BAFF signaling, thereby leading to the activation of the non-canonical NF-κB pathway via p100-to-p52 processing [25]. Our study revealed that BTLA engagement attenuated BCR/BAFF-induced Btk phosphorylation (Figure 4B), leading to the suppression of downstream signaling events, including PLCγ2 phosphorylation (Figure 4C) and p100-to-p52 processing in the non-canonical NF-κB pathway (Appendix A).

Whether the increase in BTLA expression on B cells observed in this study is a factor involved in the therapeutic response to belimumab or a protective function enhanced by belimumab remains controversial. However, a previous study reported that BTLA expression on the DN memory B cells of patients with SLE is characteristically low and not directly affected by conventional treatments, excluding belimumab [32]. As BTLA^low^ DN memory B cells are enriched in patients with active SLE and exhibit an antibody-secreting cell phenotype, belimumab possibly contributes to disease improvement by modulating the balance of BTLA-expressing memory B cells (Figure 3E).

This study has some limitations. First, this was a single-institution study, which limits the reproducibility and generalizability of our results and necessitates further validation using multiple cohorts across different institutes and countries. Furthermore, although BTLA^high^ memory B cell proportions showed an increasing trend after treatment and an inverse correlation with serum IgG levels, these relationships did not reach statistical significance, possibly due to insufficient statistical power related to the sample size. Second, whether an increase in BTLA expression on memory B cells contributes to disease improvement or whether disease improvement induces an increase in these cell subpopulations remains unclear. Therefore, further mechanistic studies should investigate the mechanisms by which belimumab treatment influences the increase in the BTLA-expressing memory B cell frequency, focusing on the B cell differentiation pathways under belimumab treatment conditions.

Nevertheless, our findings showed that belimumab treatment led to changes in the composition of memory B cells associated with the resolution of exacerbation, which may contribute to the improvement of SLE through a mechanism in which BTLA engagement attenuates BCR- and BAFF-induced B cell intracellular signaling. Our findings suggest potential inhibitory effects in addition to known BAFF inhibition and offer insights into the positioning of belimumab in future treatment strategies, considering specific cell subpopulations, such as BTLA^high^ memory B cells associated with disease activity and response to belimumab treatment.

## 4. Materials and Methods

### 4.1. Patients

Sixteen patients refractory to standard care at Juntendo University Urayasu Hospital were enrolled in this study. These patients received 10 mg/kg belimumab intravenously at weeks 0, 2, and 4 and every fourth week thereafter or 200 mg belimumab subcutaneously at weekly intervals, while continuing the standard care consisting of prednisone with one or more of the following drugs: Tacrolimus, hydroxychloroquine, azathioprine, mycophenolate mofetil, and mizoribine. All patients fulfilled the 1997 revised criteria [34] and Systemic Lupus International Collaborating Clinics criteria for SLE classification [35]. Demographic and clinical characteristics of patients are shown in Table 1. At enrollment, all patients were refractory to combination treatment with prednisone, one or more immunosuppressants, and hydroxychloroquine. Nine patients were taking hydroxychloroquine and six were taking mycophenolate mofetil. After three months, hydroxychloroquine and mycophenolate mofetil were each additionally administered to one patient. Seven, three, and three patients were taking tacrolimus, azathioprine, and mizoribine, respectively. Tacrolimus was discontinued in one patient after three months, azathioprine in one patient after 12 months, and mizoribine in one patient after six months due to decreased disease activity. All patients were taking prednisone, and prednisone doses were increased or decreased according to the disease activity. Proteinuria deterioration was observed in one patient after 12 months, and flares were controlled by increasing the mycophenolate mofetil dose. Another patient withdrew from the study because of severe side effects (*Nocardia* infection).

Blood samples were obtained from the patients before treatment and 1, 3, 6, and 12 months after belimumab treatment.

In this study, we used the samples of patients who responded favorably to belimumab treatment. Clear improvements in clinical manifestations and laboratory parameters were observed, including increased complement C4 levels (Appendix A) and decreased anti-DNA antibody titers (Appendix A), immunoglobulin (IgG and IgM) levels (Appendix A), SLEDAI-2K scores [36] (Appendix A), and steroid (prednisone) dosage (Appendix A).

Ethical approval was obtained from the Institutional Review Board of Juntendo University Urayasu Hospital (approval number: 2009-009 and U09-0001). Written informed consent was obtained from all patients prior to the study in accordance with the Declaration of Helsinki.

### 4.2. PBMC Isolation

PBMCs were isolated from the patient’s blood samples by Ficoll-Paque Plus (GE Healthcare Life Sciences, Uppsala, Sweden) density gradient centrifugation. The PBMCs were cryopreserved and used for subsequent analyses.

### 4.3. Immunofluorescence Staining and Flow Cytometry

Cryopreserved PBMCs were thawed, and cell suspensions were incubated with Hanks’ balanced salt solution containing 5% fetal bovine serum (FBS) for 30 min at 37 °C, and centrifuged at 400× *g* for 5 min. The precipitates were treated with a Fc receptor blocking reagent (Miltenyi Biotec, Bergisch Gladbach, Germany) to prevent non-specific reactions to immunostaining and sequentially stained with mouse anti-human mAbs. Anti-CD4-allophycocyanin (APC), anti-CD3-APC, anti-CD19-fluorescein isothiocyanate, and anti-CD19-phycoerythrin mAbs were purchased from Miltenyi Biotec. Anti-CD27-APC, anti-HVEM-phycoerythrin, and anti-BTLA-fluorescein isothiocyanate mAbs were purchased from BioLegend Inc. (San Diego, CA, USA). Dead cells were excluded using 7-aminoactinomycin D (BioLegend).

Stained cells were measured using FACSCalibur (BD bioscience), and data were analyzed using the FlowJo software ver 10.6.0 (BD bioscience).

### 4.4. B Cell Isolation and Culture for Flow Cytometry

PBMCs were isolated from the blood samples of healthy controls by Ficoll-Paque Plus density gradient centrifugation. B cells were negatively selected from these PBMCs using a B cell isolation kit (Miltenyi Biotec). Purified B cells were washed extensively with phosphate-buffered saline (PBS) and resuspended in the Roswell Park Memorial Institute-1640 medium (Sigma-Aldrich Japan, Tokyo, Japan) supplemented with 10% FBS. The cells were stimulated with or without anti-human IgM (10 μg/mL; Jackson Immuno Research Laboratories, West Grove, PA, USA) and recombinant human BAFF (100 ng/mL; R&D Systems, Minneapolis, MN, USA) and incubated for 18 h at 37 °C in 5% CO_2_. After incubation, the cells were washed with PBS, centrifuged at 400× *g* for 5 min, and used for flow cytometry as previously described.

### 4.5. Sodium Dodecyl Sulfate–Polyacrylamide Gel Electrophoresis and Western Blotting

Isolated B cells were incubated with the anti-BTLA mAb (10 μg/mL; BioLegend) or isotype controls (anti-IgG2a mAb: 10 μg/mL; BioLegend) for 20 min on ice. Next, the cells were washed and resuspended in the Roswell Park Memorial Institute-1640 medium supplemented with 10% FBS, incubated for 5 min at 37 °C, cross-linked with goat anti-mouse IgG (20 μg/mL, Invitrogen), and stimulated with or without anti-human IgM (10 μg/mL) combined with human BAFF (100 ng/mL) for 1 min or 4 h. The cells were washed with PBS and lysed in a lysis buffer (50 mM Tris-HCl [pH 8.0], 150 mM NaCl, and 0.5% NP-40) with phosphatase inhibitors (Thermo Fisher Scientific, Waltham, MA, USA) and protease inhibitors (Roche, Rotkreuz, Switzerland) for 20 min on ice. The lysates were centrifuged at 20,000× *g* for 20 min, and the supernatants were collected. Subsequently, the cell lysates (5 μg) were separated by sodium dodecyl sulfate–polyacrylamide gel electrophoresis and transferred onto polyvinylidene fluoride membranes. The membranes prepared from 1 min-stimulated samples were probed with antibodies against phospho-Syk (Tyr^352^), Syk, phospho-Btk (Tyr^551^), Btk, phospho-PLCγ2 (Tyr^1217^), and PLCγ2. In contrast, the membranes prepared from 4 h-stimulated samples were probed with an antibody against NF-κB2 (p100/p52) subunit. In both cases, glyceraldehyde-3-phosphate dehydrogenase served as a cytoplasmic protein loading control. The membranes were subsequently incubated with horseradish peroxidase-conjugated rabbit anti-mouse IgG or goat anti-rabbit IgG (Dako, Glostrup, Denmark) secondary antibodies. Specific bands were detected using the Amersham Imager 600 (GE Healthcare, Chicago, IL, USA) and analyzed using the ImageJ software ver 1.54g(National Institutes of Health, Bethesda, MD, USA) to quantify the band intensities. Antibodies against phospho-Syk, Syk, phospho-Btk, Btk, phospho-PLCγ2, PLCγ2, and NF-κB2 p100/p52 were purchased from Cell Signaling Technology Inc. (Danvers, MA, USA), and glyceraldehyde-3-phosphate dehydrogenase mAbs were purchased from Sigma-Aldrich (St. Louis, MO, USA).

### 4.6. Statistical Analyses

All statistical analyses were conducted using the GraphPad Prism software 10 (GraphPad, San Diego, CA, USA). Mixed-effects model followed by Dunnett’s post hoc test was used to compare the changes in cell subpopulations, clinical parameters, and laboratory data in belimumab-treated patients with SLE over time. Each post-treatment value was compared to the corresponding value before belimumab treatment.

The strength of the associations between specific cell subpopulations and clinical and laboratory parameters was evaluated using Pearson’s or Spearman’s rank correlation coefficients. Two-way analysis of variance followed by Tukey’s multiple comparison test was used to compare the signaling protein phosphorylation levels. Statistical significance was set at *p* < 0.05.

## Figures and Tables

**Figure 1 ijms-26-11323-f001:**
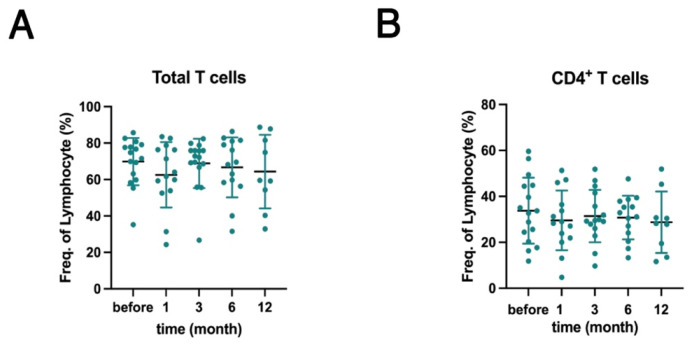
Belimumab treatment did not affect the T cell subpopulations in patients with systemic lupus erythematosus (SLE). Fluorescence-activated cell sorting (FACS) analysis was performed on T cell subsets within the peripheral blood mononuclear cells (PBMCs) isolated from patients with SLE before and 1, 3, 6, and 12 months after belimumab treatment. (**A**) Frequencies of total (CD3^+^) and (**B**) CD4^+^ (CD3^+^ CD4^+^) T cells expressed as percentages of total PBMCs. Each dot in the plot represents an individual sample (*n* = 16), and bars represent the mean ± standard deviation (SD). Statistical analyses were performed using a mixed-effects model followed by Dunnett’s post hoc test to compare the data before and 1, 3, 6, and 12 months after treatment.

**Figure 2 ijms-26-11323-f002:**
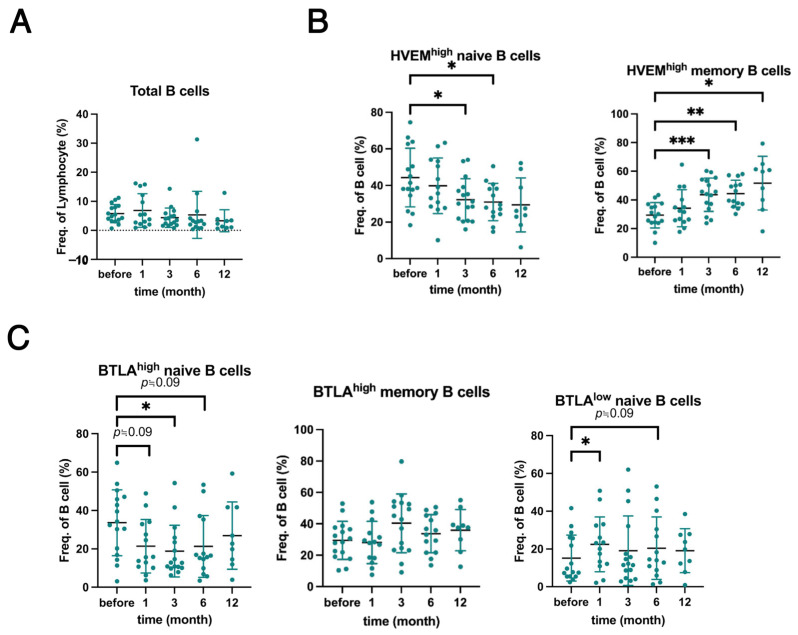
Belimumab treatment affects the B-cell subpopulations in patients with SLE. FACS analysis was performed on B cell subsets within the PBMCs isolated from patients with SLE before and 1, 3, 6, and 12 months after treatment with belimumab. (**A**–**C**) Frequencies of all cell subsets: (**A**) Total B cells (CD3^−^CD19^+^), (**B**) HVEM^high^ naive B cells (CD19^+^CD27^−^HVEM^high^), HVEM^high^ memory B cells (CD19^+^CD27^+^HVEM^high^), (**C**) BTLA^high^ naive B cells (CD19^+^CD27^−^BTLA^high^), BTLA^high^ memory B cells (CD19^+^CD27^+^BTLA^high^), and BTLA^low^ naive B cells (CD19^+^CD27^−^BTLA^low^). Each dot represents an individual patient sample (*n* = 16), and bars represent the mean ± SD. Statistical analyses were performed using a mixed-effects model followed by Dunnett’s post hoc test to compare the data before and 1, 3, 6, and 12 months after treatment. Statistically significant changes are marked with asterisks (* *p* < 0.05, ** *p* < 0.01, and *** *p* < 0.001).

**Figure 3 ijms-26-11323-f003:**
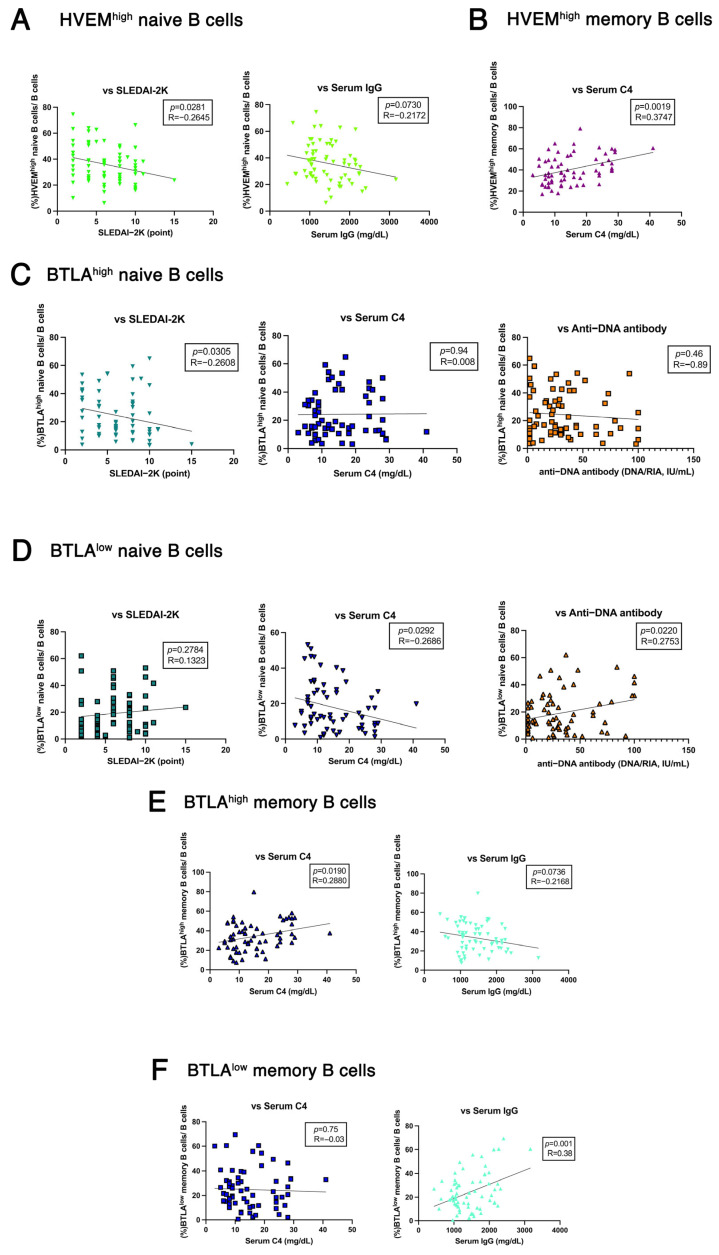
B- and T-lymphocyte attenuator (BTLA)-positive B cells contribute to improved disease activity in SLE. Correlation analysis between cell subpopulations analyzed via FACS and clinical parameters, including serologic data and SLE disease activity index 2000 (SLEDAI-2K) score, of patients with SLE. (**A**) HVEM^high^ naive B cell frequency vs. SLEDAI-2K scores and serum IgG levels. (**B**) HVEM^high^ memory B cell frequency vs. serum complement C4 levels. (**C**) BTLA^high^ naive B cell frequency vs. SLEDAI-2K scores, serum complement C4 levels, and anti-DNA antibody titers. (**D**) BTLA^low^ naive B cell frequency vs. SLEDAI-2K scores, serum complement C4 levels, and anti-DNA antibody titers. (**E**) BTLA^high^ memory B cell frequency vs. serum complement C4 and IgG levels. (**F**) BTLA^low^ memory B cell frequency vs. serum complement C4 and IgG levels. Each symbol represents an individual sample, and lines represent the regression lines. Pearson’s correlation analysis was performed for most statistical analyses, except for correlations with SLEDAI-2K scores, which were analyzed using Spearman’s rank correlation analysis. *p* and *R* represent the *p* values and the correlation coefficient (Pearson’s or Spearman’s rank), respectively.

**Figure 4 ijms-26-11323-f004:**
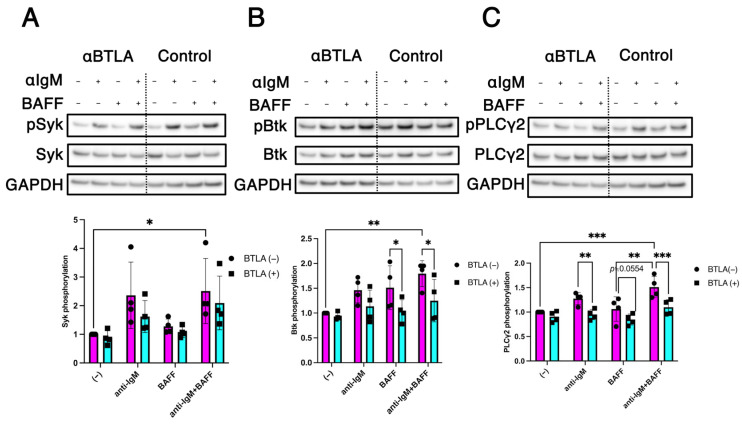
BTLA engagement suppresses B cell receptor (BCR) and B cell-activating factor (BAFF) co-stimulation-induced Bruton tyrosine kinase (Btk) and phospholipase C-gamma 2 (PLCγ2) phosphorylation. (**A**–**C**) Isolated B cells from healthy controls were incubated with the anti-BTLA monoclonal antibody (mAb; αBTLA) or isotype controls. BTLA was crosslinked and stimulated with or without anti-human IgM (αIgM) and human BAFF. Levels of signaling proteins were determined via western blotting. Glyceraldehyde-3-phosphate dehydrogenase (GAPDH) expression was used as a loading control. Upper panels show the representative blots for (**A**) phospho-Syk (pSyk) and total Syk, (**B**) phospho-Btk (pBtk) and total Btk, and (**C**) phospho-PLCγ2 (pPLCγ2) and total PLCγ2. Lower panels show the results of densitometric analysis of the band intensities in the upper panels. Fold changes in Syk, Btk, and PLCγ2 phosphorylation levels were calculated as the ratios of pSyk:Syk, pBtk:Btk, and pPLCγ2: PLCγ2 band intensities, respectively, and normalized to GAPDH. Assays were performed in quadruplicate. Each dot or square represents an individual experiment, and bars represent the mean ± SD. Two-way analysis of variance (ANOVA) and Tukey’s multiple comparisons test were used to analyze the results. Statistically significant changes are marked with asterisks (* *p* < 0.05, ** *p* < 0.01, and *** *p* < 0.001). Numbers on the graph indicate the *p* values.

**Table 1 ijms-26-11323-t001:** Patient characteristics.

Characteristics	Patients
Sex; *n* = 16	
Female; *n* (%)	13 (81.2%)
Age (years); M (IQR); *n* = 16	41 (32.0–49.8)
SLE disease duration (years); M (IQR); *n* = 16	12 (7.5–19.0)
SLEDAI-2K; M (IQR); *n* = 16	6 (4.8–8.5)
Prednisone equivalent dose (mg/day); M (IQR); *n* = 16	10 (7.3–13.1)
Previous exposure to corticosteroids (years); M (IQR); *n* = 16	12 (7.5–19.0)
Refractory manifestations; *n* (%)	
Musculoskeletal	12 (75.0%)
Mucocutaneous	14 (87.5%)
Renal	12 (75.0%)
Administration methods; *n* (%)	
Intravenously	1 (6.2%)
Subcutaneously	15 (93.7%)

SLE, systemic lupus erythematosus; SLEDAI-2K, systemic lupus erythematosus disease activity index 2000; M, median; IQR, interquartile range.

## Data Availability

Raw data supporting the conclusions of this study are available upon request from the authors.

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
