# Peer review of "BTLA-Expressing Memory B Cells Are Associated with Belimumab-Induced Improvement in Systemic Lupus Erythematosus"

_ijms, 2025, doi:10.3390/ijms262311323_

Round 1
Reviewer 1 Report
Comments and Suggestions for Authors
Here are some points for the Authors to consider.
- Paired t-tests is inappropriate statistical approach when multiple groups exist. Instead, ANOVA or non-parametric equivalent test with post-hoc groupwise comparisons should be performed.
- What is the expression of HVEM and BTLA on peripheral blood Age-associated B-cells which are implicated in SLE pathogenesis?
- The mechanistic studies are interesting, however it is difficult to judge the presented differences (eg in pBtk) at the western blots. To me, levels seem to be comparable between aBTLA and Control groups. Same for the non-canonical NF-kb pathway shown in the supplement.
- Supplementary material. There is some confusion with the legends as they do not fit with the presented data.
Author Response
- Paired t-tests is inappropriate statistical approach when multiple groups exist. Instead, ANOVA or non-parametric equivalent test with post-hoc groupwise comparisons should be performed.
Response1: Thank you for this important observation. We agree that paired t-tests were inappropriate for our multiple group comparisons. We have revised the statistical analyses of Figure 1, 2 and Figure S3 by performing a Mixed-effects model followed by Dunnett’s post hoc test for multiple comparisons against the control group. All affected figures and their legends have been updated accordingly.
- What is the expression of HVEM and BTLA on peripheral blood Age-associated B-cells which are implicated in SLE pathogenesis?
Response2: In this study, we were unable to analyze the expression of HVEM and BTLA on age-associated B cells (ABCs) in SLE patients, as our facility only had access to a FACSCalibur instrument capable of measuring up to four parameters. Therefore, such data could not be obtained.
To our knowledge, there have been no published reports directly describing the expression of HVEM and BTLA on ABCs in SLE patients. Previous studies have shown that ABCs can differentiate from naive B cells following BCR stimulation, cooperating with Toll-like receptor 7 or 9 agonists and exposure to interferon-γ or interleukin-12 (Hao, Y et al. Blood 2011, 118, 1294-1304) (Cancro, M. P. et al. Annu. Rev. Immunol 2020, 38, 315-340) following BCR stimulation and exhibit constitutive BCR activation (Imabayashi et al. Sci Adv 2025, 11, (16), eadt8199). In SLE, BTLA expression levels have been reported to be decreased on naive and double-negative (CD27-IgD-) memory B cells (Wiedemann, A et al. Front. Immunol 2021, 12, 667991). Additionally, our in vitrodata showed a tendency for HVEM expression to increase and BTLA expression to decrease upon BCR activation (Figure S1A, B). Therefore, although direct data for ABCs are lacking, it is plausible that ABCs, given their constitutive BCR activation, ABCs may exhibit slightly higher levels of HVEM expression and lower levels of BTLA expression compared to other B cell subsets in SLE patients. Further investigation will be needed to confirm this hypothesis.
- The mechanistic studies are interesting; however, it is difficult to judge the presented differences (eg in pBtk) at the western blots. To me, levels seem to be comparable between aBTLA and Control groups. Same for the non-canonical NF-kb pathway shown in the supplement.
Response3:
For the pBtk:
We apologize for the difficulty in interpretation caused by variability in GAPDH normalization across lanes; the bands shown Figure 4B represent the clearest images available from this experiment. To provide more quantitative and transparent assessment complementing Figure 4B, we report the mean band intensity of pBtk for each group in the table below (Appendix Table 1), calculated as the average across four independent blots (n = 4).
|
|
αBTLA |
Control |
||||||
|
αIgM (stimulation) |
− |
+ |
− |
+ |
− |
+ |
− |
+ |
|
BAFF (stimulation) |
− |
− |
+ |
+ |
− |
− |
+ |
+ |
|
pBtk (Mean) |
16893 |
19548 |
19551 |
21241 |
18198 |
21298 |
18642 |
20300 |
|
Btk (Mean) |
10418 |
10237 |
11749 |
11567 |
12355 |
11124 |
11373 |
9577 |
|
GAPDH (Mean) |
12470 |
12359 |
11933 |
11038 |
10452 |
9386 |
8098 |
8635 |
|
Normalized values used for graph (mean) |
0.93 |
1.13 |
1.04 |
1.24 |
1 |
1.46 |
1.51 |
1.79 |
For the NF-κB:
Upon careful re-examination, we believe that there is no substantial discrepancy between visual appearance and quantitative data for p100 (Figure S2).
After receiving the reviewer’s comment, we re-examined our data and found that we had presented an incorrect graph for p52 in Figure S2. We apologize for this mistake and have now revised the graph with the corrected version.
Based on the revised graph, there was a trend toward increased p52 levels following co-stimulation with anti-IgM and BAFF, and this effect appeared to be suppressed by BTLA engagement (Figure S2). This finding remains unchanged from those observed in the previous version of Figure S2. Although p52 levels are difficult to assess reliably from visual appearance alone, we have included comprehensive quantified band intensity data (all available data; n =2) with Appendix Figure 1 for objective assessment. Given the limited sample size, these densitometric values should be interpreted with caution. Although they do not affect our overall conclusions, we believe they may offer supplementary support for the findings presented in this study. Furthermore, we have clarified the relevant statements in the Results section to reflect the corrected data, and to avoid any misinterpretation caused by the previous wording. We believe these revisions ensure greater transparency and accuracy in our manuscript.
Appendix Figure 1 (legend)
Western blot analysis showing protein bands for p52 and GAPDH from two independent experiments (n = 2). The blot image used for Figure S2 is from the “Blot 1”. Quantitative values shown below each band represent individual measurements of p52 normalized to GAPDH and expressed as ratio relative to the negative control. GAPDH values indicate the raw (unnormalized) band intensities. All experimental data used for the generation of Figure S2 are presented.
- Supplementary material. There is some confusion with the legends as they do not fit with the presented data.
We appreciate the points about the mismatch between some legends and the corresponding data in the Supplementary materials. We have corrected the legends to accurately match the figures and have updated the Supplementary material accordingly.
Reviewer 2 Report
Comments and Suggestions for Authors
The article is clear and adequately describes the experiments and results. Therefore, it could be accepted for publication after considering some suggestions regarding statistical analysis.
The statistical analysis in Figure 1 and Figure 2 is not correct. Paired t-tests were used; however, every time a t-test is performed, there is a probability of committing a type I error. If several tests are performed, these probabilities accumulate, so a correction must be applied or an appropriate statistical test must be used.
Figure 1 and lines 100-101 indicate that there was a reduction in lymphocyte frequency; however, a p=0.08 is not significant.
Figure 2 indicates that p=0.06 is significant, so the reference value to be considered significant must be clarified.
Pearson's correlation is used in the correlation analysis; however, the SLEDAI usually behaves like a non-normal ordinal or semi-continuous scale, so Spearman's test is more appropriate.
Author Response
Thank you very much for your instructive comments and suggestions. We deeply appreciate your support.
The statistical analysis in Figure 1 and Figure 2 is not correct. Paired t-tests were used; however, every time a t-test is performed, there is a probability of committing a type I error. If several tests are performed, these probabilities accumulate, so a correction must be applied or an appropriate statistical test must be used.
Figure 1 and lines 100-101 indicate that there was a reduction in lymphocyte frequency; however, a p=0.08 is not significant.
Figure 2 indicates that p=0.06 is significant, so the reference value to be considered significant must be clarified.
Pearson's correlation is used in the correlation analysis; however, the SLEDAI usually behaves like a non-normal ordinal or semi-continuous scale, so Spearman's test is more appropriate.
Response: We agree that paired t-tests were inappropriate for our muti-group comparisons. We have revised the statistical analyses of Figure 1, 2 and Figure S3 by performing a Mixed-effects model followed by Dunnett’s post hoc test for multiple comparisons against the control group. All affected figures and their legends have been updated accordingly. Statistical significance was set at p < 0.05, as described in the Materials and Methods (lines 460-461).
For the association analyses between cell subpopulations and SLEDAI-2K scores, we revised the statistical method to Spearman’s rank correlation coefficients, as suggested, considering the ordinal nature of the SLEDAI-2K scores.
Round 2
Reviewer 1 Report
Comments and Suggestions for Authors
I am content with the revised version.